# Satellite IoT Edge Intelligent Computing: A Research on Architecture

**Junyong Wei [1,2]** **, Jiarong Han [1,2] and Suzhi Cao [2,\***

[1]   University of Chinese Academy of Sciences, Beijing 100049, China; weijunyong17@mails.ucas.ac.cn (J.W.);
     hanjiarong18@mails.ucas.ac.cn (J.H.)
[2]   Key Laboratory of Space Utilization, Technology and Engineering Center for Space Utilization, Chinese
     Academy of Sciences, Beijing 100094, China
\*   Correspondence: caosuzhi@csu.ac.cn; Tel.: +86-010-8217-8226

**Abstract:** As the number of satellites continues to increase, satellites become an important part of the IoT and 5G/6G communications. How to deal with the data of the satellite Internet of Things is a problem worth considering and paying attention to. Due to the current on-board processing capability and the limitation of the inter-satellite communication rate, the data acquisition from the satellite has a higher delay and the data utilization rate is lower. In order to use the data generated by the satellite IoT more effectively, we propose a satellite IoT edge intelligent computing architecture. In the article, we analyze the current methods of satellite data processing, combined with the development trend of future satellites, and use the characteristics of edge computing and machine learning to describe the satellite IoT edge intelligent computing architecture. Finally, we verify that the architecture can speed up the processing of satellite data. By demonstrating the performance of different neural network models in the satellite edge intelligent computing architecture, we can find that the lightweight of neural networks can promote the development of satellite IoT edge intelligent computing architecture.

**Keywords:** IoT; satellite internet of things; edge computing; edge intelligent computing; deep learning

## 1. Introduction

The Internet of Things (IoT) is one of the new directions for the future development of the Internet, and intelligent networking devices will be connected [1,2]. Although the number of IoT devices is increasing, some remote areas do not have network coverage. Terrestrial wireless networks are only covered by 20% due to extreme terrain or communication distance and cost constraints [3]. The wide-area coverage of satellite radio networks can solve the problem of ground coverage and solve the problem of communication interruption caused by natural disasters. Obviously, satellites have become an important part of the Internet of Things [4,5]. At the same time, satellites are an achievable and powerful complement to terrestrial networks and future 5G/6G communications [6–8].

The number of remote sensing satellites has increased dramatically in satellite launches in recent years [9,10]. After acquiring the remote sensing image data of the satellite, the researchers used the artificial intelligence algorithm and the powerful computing power of the ground data center to extract the hidden information in the remote sensing image. However, researchers need to spend a lot of time and cost to complete this process. In the existing satellite communications, most of the observation, relay, and communication satellites are single-star and single-chain, and there is no network. Due to the limitation of energy consumption, the available processors on the satellites have poor performance and cannot meet the growing demand for space computing tasks [11]. At the same time, the satellite communication rate between satellites and other satellites, and between satellites and the ground, is generally not improved. The amount of data generated by the on-board sensor is large, causing a high

delay in the data transmission process, which is very disadvantageous for scenes with high real-time requirements (such as early warning).

With the advancement of aerospace technology, commercial satellite companies such as OneWeb, O3b [12], and SpaceX Starlink [13] have proposed satellite constellation plans, increasing investment and research on satellite-related technologies, making satellites more powerful and smarter. The computing power of the high-performance computing unit mounted on the satellite has been greatly improved. Real-time intelligent computing of satellite remote sensing images will become possible.

Compared with the method of transmitting satellite image data to the satellite ground station cloud computing data center and using deep learning for centralized processing, this paper proposes a method using edge intelligent computing processing and describes the satellite IoT edge intelligent computing architecture. Combined with edge computing technology [14], such as mobile edge computing [15,16], fog computing [17,18] fully utilizes the computing power from the cloud to the edge of the network to decompose the deep-learning tasks. Here, we use the more powerful on-board computing power in the future to place the deep-learning algorithm model at the source of satellite data generation to achieve edge intelligent computing. This architecture can reduce the amount of data transmitted from satellites, reduce data processing and communication delays, improve the bandwidth utilization of inter-satellite links, and reduce the pressure on data processing of satellite ground stations.

This paper is organized as follows. Section 2 describes the related research work of satellite Internet of Things and edge intelligent computing and distributed deep learning. Then, Section 3 introduces the significance of edge computing and deep learning in satellite IoT and how to implement satellite IoT edge intelligent computing architecture. Section 4 mainly verifies the proposed solution, including the connectivity and coverage performance of the satellite IoT and the performance of the edge intelligent computing architecture. Finally, Section 5 concludes this work.

## 2. Related Work

### 2.1. Related Research on Satellite Internet of Things

The role of satellites in the IoT is irreplaceable and related technologies of satellite IoT have become one of the hot research topics. In [4], the authors outline the IoT architecture based on the LEO satellite constellation, including LEO satellite constellation structure, efficient spectrum allocation, heterogeneous networks compatibility, and access and routing protocols. The importance of using satellite communication systems to the Internet of remote Things (IoRT) is described in [5]. The research includes MAC protocols for satellite-routed sensor networks, efficient IPv6 support, quality of service (QoS) management, heterogeneous networks interoperability, and group-based communications. Article [19] introduces SDN/NFV and microsatellite technology into the space satellite IoT and evaluates the impact of different orbital configurations and carrier frequencies on data rates, link latencies, and next-hop availability and access durations.

### 2.2. Related Research on Distributed Deep Learning

In recent years, many companies have released their own distributed machine-learning models, including distributed training models and distributed reasoning models [20–27]. A distributed training system architecture called Parameter Server is proposed in the paper [21]. In this architecture, the nodes are divided into two categories: the Parameter Server node is responsible for storing and managing parameters; the worker node is responsible for performing specific calculation tasks, based on part of the training data, and obtaining the gradient of the parameters. Ring Allreduce [22] is a very mature and efficient algorithm in the field of high-performance computing (HPC), but it is rarely used in distributed machine learning. In recent work, Uber's open-source distributed deep-learning framework, Horovod [23], and Baidu's open-source deep-learning framework, PaddlePaddle [24], have introduced the Ring Allreduce algorithm to accelerate the training process of deep neural networks. At

present, Alibaba open-source is the first lightweight deep-learning end-side inference engine [25]. The engine can directly run the deep-learning model on an embedded device that is versatile, lightweight, high-performance, and easy to use.

### 2.3. Related Research on Edge Intelligent Computing

Edge intelligent computing combines edge computing with deep-learning technology, which is mainly used in the field of image processing [28–32]. Article [28] proposes Edgent, a collaborative and on-demand deep neural network (DNN) collaborative reasoning framework with device edge synergy. DNN reasoning is accelerated by early exit at the appropriate intermediate DNN layer to further reduce computational delay. An offloading strategy in the edge intelligent computing environment is designed to optimize the deep-learning performance of the Internet of Things in [29]. An energy-saving communication based on deep-learning-based edge computing is proposed in [30]. The authors have designed and implemented an energy-efficient IoT camera named CamThings, which reduces power consumption by transmitting only images of interest classified using edge computing. In paper [31], the author expounds the importance of edge intelligence, summarizes the challenges and opportunities brought by edge intelligence, and demonstrates the implementation of convolutional neural network based (CNN-based) objects on embedded edge devices on unmanned aerial vehicles (UAVs). Similarly, the article [32] emphasizes the importance of edge intelligence and discusses the future challenges of artificial intelligence algorithms running on IoT devices at the edge of the network under power constraints.

## 3. Satellite IoT Edge Intelligent Computing Architecture

The edge intelligent computing of the satellite IoT is mainly to apply edge computing and deep learning technology to the satellite IoT. Next, we will introduce the principles of the satellite IoT edge intelligent computing architecture. This chapter includes satellite IoT edge computing, distributed intelligent computing architecture in satellite IoT edge computing.

### 3.1. Satellite IoT Edge Computing

The satellite Internet of Things can take advantage of emerging network technologies on the ground. In recent years, there have been studies on satellite IoT edge computing, considering the satellite's ever-increasing computing and storage capabilities, treating each satellite as an edge node, enabling on-orbit processing and minimizing the delay caused by satellite transmission [33–35].

This paper describes a cloud-edge layered satellite IoT edge computing paradigm. The hierarchical network of satellite IoT also needs to be supported by terrestrial data centers. Specifically, the cloud-edge stratified satellite IoT edge computing system consists of three parts: the satellite IoT cloud node, the satellite IoT edge node, and the ground data center. Figure 1 shows a cloud-edge layered satellite IoT edge computing system.

The satellite IoT edge nodes have computing and storage capabilities and use a common virtualization platform that can deploy different services as needed. Satellite IoT edge nodes can communicate with each other, and satellite IoT edge nodes and satellite IoT cloud nodes can cooperate with each other. This can bring two benefits. First, satellite IoT edge nodes can request assistance from satellite IoT cloud nodes or ground data centers to offload their computing tasks to them. Second, satellite IoT edge nodes can also accept tasks from satellite IoT cloud nodes or terrestrial data centers, or establish fast service clusters with other satellite IoT edge nodes.

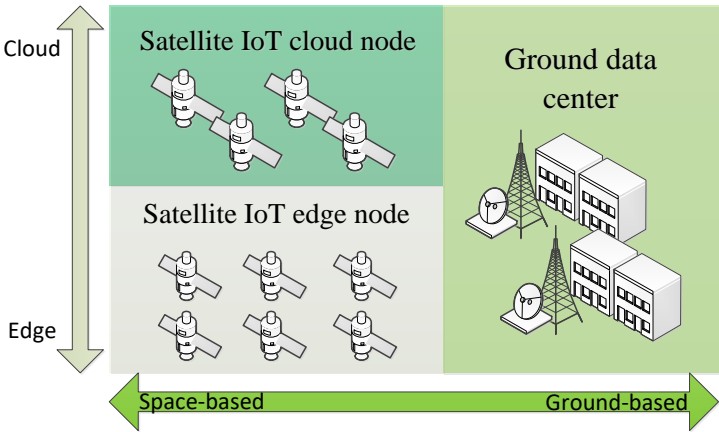

**Figure 1.** Cloud-edge layered satellite IoT edge computing system.

The satellite IoT cloud node assumes the function of the satellite IoT data center, which is equivalent to the aggregation node of the satellite IoT edge node. Similarly, they can communicate with each other and connect the satellite IoT edge nodes to the ground data center. Satellite IoT cloud nodes have more powerful computing and storage capabilities than satellite IoT edge nodes. Satellite IoT cloud nodes are equipped with heterogeneous resources, such as CPU, GPU, and FPGA. It can not only handle various applications unloaded from satellite IoT edge nodes, but also complete task scheduling, task analysis, data fusion, intelligent distribution, and fast service cluster construction of the entire satellite network.

The satellite IoT edge node and the satellite IoT cloud nodes are logically layered. But in reality, they won't be highly layered in space. Satellite IoT cloud nodes can operate on the same orbit as the satellite IoT edge nodes, or on geosynchronous orbits. It is similar to a mobile base station and acts as a group leader for a particular area, managing satellite IoT edge nodes. The satellite IoT edge node works in conjunction with the satellite IoT cloud node to maintain the normal operation of the satellite IoT.

The ground data center has the capability of a large cloud computing center that can communicate with satellite IoT nodes or the ground Internet. Compared to satellite IoT nodes, the ground data centers have the highest computing power and the most storage resources.

The satellite IoT edge node, the satellite IoT cloud node, and the ground data center constitute a three-layer computing architecture of the cloud-edge layered satellite IoT edge computing system. From the edge to the cloud, the computing power of each layer is gradually increasing. The task strength of each layer is as follows. Low-complexity computing can be performed on satellite IoT edge nodes. High-complexity and high real-time computing are suitable for implementation in satellite IoT cloud nodes. When the mission strength exceeds the satellite IoT's affordability, the mission will be passed to the ground data center.

In particular, satellite IoT edge nodes and satellite IoT cloud nodes can implement network slicing through SDN/NFV technology. The data layer and the control layer of the satellite IoT are separated. The data layer can reflect the state of the current network, and the control layer can allocate network resources to implement network slicing. Network slicing can provide independent network instances by satisfying the differentiated requirements of different service qualities, such as bandwidth and delay. At the same time, it can enhance the flexibility and adaptability of the network.

### 3.2. Distributed Intelligent Computing Architecture in Satellite IoT Edge Computing

Satellite IoT edge computing is the basis for edge intelligence computing for the satellite IoT. Currently, the training and reasoning required for deep-learning models has been deployed in terrestrial cloud computing data centers. In the future, satellite IoT sensors have a large amount of image data to inference every day. And we know that the cost of image inference requires up to gigabit floating point

arithmetic. If all the image data are moved to the ground data center for reasoning, the calculation and storage pressure of the ground cloud computing center will increase greatly.

Due to the improvement of the hardware level, the computing and storage capabilities of the satellite nodes are greatly improved. While the satellite local control is satisfied, some remaining computing and storage capabilities are not fully exploited. When there are remaining computing and storage resources, the satellite IoT edge nodes can cooperate with the satellite IoT cloud nodes, take advantage of satellite edge computing to undertake a certain amount of deep-learning tasks. Therefore, the extension of cloud intelligence to edge intelligence has become an inevitable trend. In general, satellite edge intelligent computing is the analysis of data at the source of satellite sensors rather than sending data to the ground cloud for analysis. It will become a key driver for the realization of intelligent satellite IoT.

### 3.2.1. Cross-Layer Satellite IoT Edge Intelligent Computing Architecture

At present, the scale of problems in machine learning and deep learning is getting larger and larger, and the amount of data and the number of parameters involved are also rising sharply. Traditional single-machine computing has difficulty providing enough computing power and storage resources. Therefore, distributed model training and inference are gradually becoming mainstream. Unlike single-machine training, distributed machine learning involves more details that need to be carefully considered, including global sharing of parameters and the impact of single-node performance "short boards" on cluster performance. In addition, in practical industrial applications, the scale of parameters generated by the data of the order of 1 TB to 1 PB during the training process is between $10^9$ and $10^{12}$ [21]. Parameter sharing has become a difficult point.

Using the idea of distributed deep-learning training, in the satellite IoT edge intelligent computing architecture, the satellite IoT edge node near the data side can be used as the worker node to perform specific computing tasks. The advantage of this is that, in satellite-like remote-sensing observations and the like, the satellite IoT edge node is both a worker node and a data source. This way we can easily get a natural distributed data set. Therefore, each satellite IoT edge node in the distributed cluster can directly use the data collected by them for training, which saves the network bandwidth consumed by data acquisition. At the same time, satellite IoT cloud nodes that are not directly connected to the data source but have larger storage resources and higher computing power are the nodes responsible for storing and managing parameters. At the same time, satellite IoT cloud nodes with larger storage resources and more computing power are used as nodes responsible for storage and management parameters. Two types of nodes use inter-satellite links for communication and data transmission.

In addition, considering the constraints of computing power and the constraints of power consumption of embedded devices on satellites, the neural network model must be subdivided or simplified. A deep neural network can be thought of as a directed graph with multiple network layers under each image. If the entire neural network model is run directly on the edge computing device, performance may be poor. We can design an appropriate lightweight neural network by weighing the relationship between inferential accuracy and latency. At the same time, in the satellite IoT scenario, we can store the corresponding pretraining model on the satellite IoT cloud node to accelerate the process of model training. In this process, there are two ways to obtain the pretraining model: firstly, it can be uploaded by the ground station and stored persistently in the satellite IoT cloud node; secondly, the satellite IoT cloud node can be fully utilized to obtain the pretraining model. The neural network training process of the cross-layer satellite IoT edge intelligent computing architecture is depicted in Figure 2.

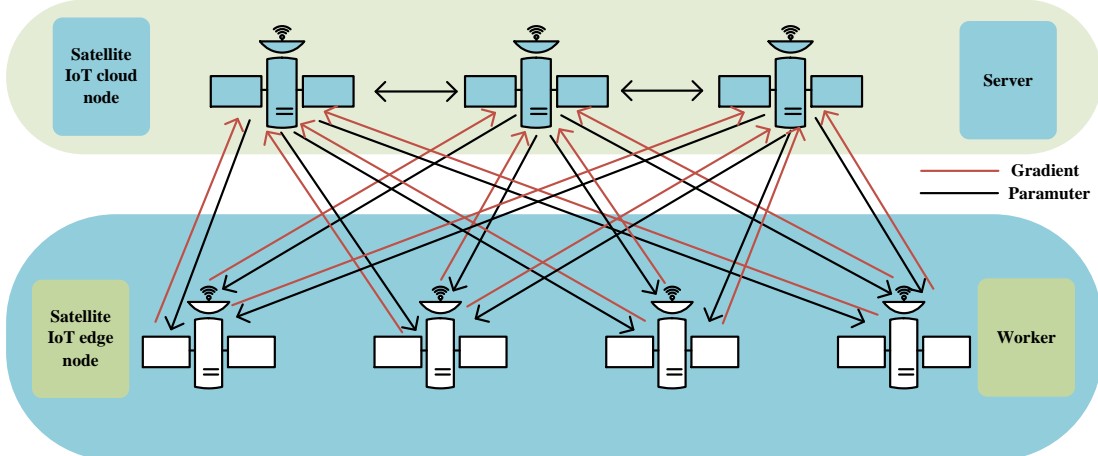

**Figure 2.** Cross-layer satellite IoT edge intelligent computing architecture's neural network training process.

After the computing task is offloaded to the satellite cluster, the satellite IoT edge node first obtains the pretraining parameters of the model from the satellite IoT cloud nodes, and then uses the data collected by itself to optimize the model, calculate the parameter gradient, and send it to the satellite IoT cloud nodes. After that, the satellite IoT cloud nodes aggregate the gradients, and the parameters are updated and broadcast to each worker nodes. In addition to performing training tasks, inference tasks can also be performed in a space-based decentralized computing architecture. In addition to performing training tasks, inference tasks can also be performed in the satellite IoT edge intelligent computing architecture. Each satellite IoT edge node contains the latest model, which can directly infer the collected data and feed the result back to the ground data center or client.

### 3.2.2. Training-Inference-Isolated Satellite IoT Edge Intelligent Computing Architecture

A tricky problem in the edge intelligent computing architecture like the "master–slave" structure above is the network communication consumption during the training process. The satellite IoT cloud nodes, as the core of parameter gradient collection and processing, need to establish communication links with each satellite IoT edge node. Therefore, with the expansion of the scale of participating computing nodes, the communication cost will also rise sharply, which is a big challenge for satellite networks. Therefore, for a task with a large amount of computation and a large number of participating nodes, we need a distributed intelligent computing architecture with less communication cost.

In the satellite IoT edge intelligent computing architecture, the algorithm can be used to alleviate the network communication pressure caused by the parameter-sharing process, as shown in Figure 3. Specifically, we chose to deploy a deep-learning framework on a cluster of satellite IoT cloud nodes with more computing power. In this framework, each satellite IoT cloud node is abstracted into a node in a logical ring, participating in both parameter calculation and parameter storage. Each node in the logical ring receives data from its left neighbor and sends data to its right neighbor.

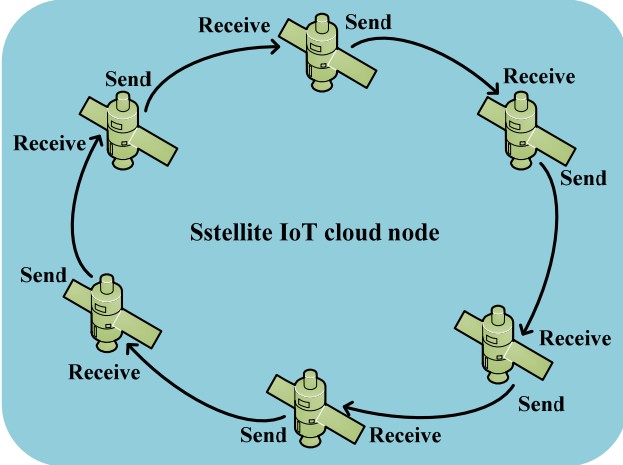

**Figure 3.** Training-inference-isolated satellite IoT edge intelligent computing architecture.

In the training process of the neural network, each round of training can be divided into two stages. First, in the gradient calculation stage, each satellite IoT cloud node receives the data collected or generated by the edge node as its own training set and performs the calculation task to get the gradient of the parameter. The second stage is the parameter update phase, in which each node shares its own calculation results with other nodes, and finally each node has the final updated parameters. Referring to the Ring Allreduce algorithm, this process is divided into two steps: in the first step, gradient accumulation is performed, and then the gradient calculated by each node is divided into $N$ segments ($N$ is the number of nodes in the logical ring). And each node follows the rules to pass only one gradient segment to the adjacent node. After $N$-1 rounds, every segment of gradient has been accumulated. Finally, these segments of gradient are distributed on different nodes. In the second step, each node exchanges its own final gradient segment and accumulates the gradient synchronization of all nodes to update the parameters. The training process of the training-inference-isolated satellite IoT edge intelligent computing architecture is shown in Figure 4.

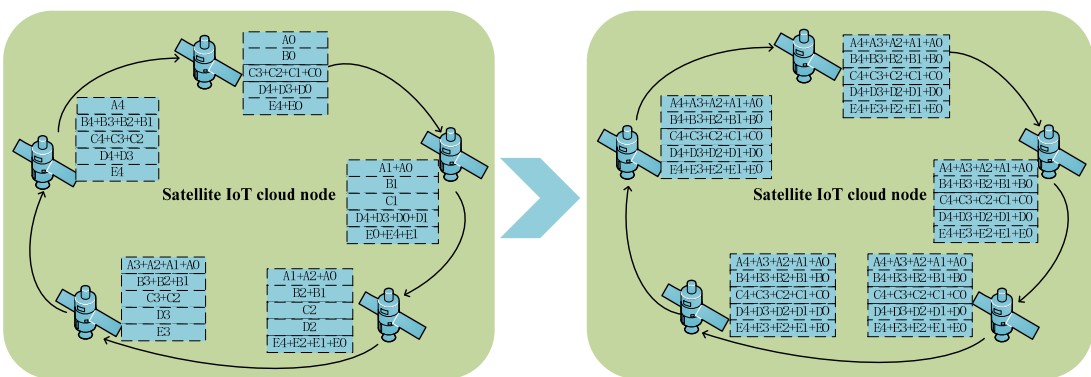

**Figure 4.** Training-inference-isolated satellite IoT edge intelligent computing architecture's neural network training process.

In this architecture consisting of $N$ satellites, in step one, each satellite will send $N - 1$ data segments, and in step two, each satellite will receive $N - 1$ data segments. Further, the amount of data involved in each data transfer is $K/N$ ($K$ is the total number of values that each data segment adds on a different satellite). Therefore, we can calculate the total amount of data involved in the communication with the following formula:

$$Data_{Tran} = 2(N-1)\frac{K}{N}$$

In the satellite IoT edge intelligent computing architecture, we envision offloading the inference task to the satellite IoT edge node. After the satellite IoT edge node obtains the trained neural network from the satellite IoT cloud node, each satellite can infer the received data, and the result is fed back to the ground data center or the user end. At this time, the model parameters on each satellite are the same. When there is an inference task, the satellite can select local processing or offload the task to nearby satellites for distributed collaborative reasoning according to the amount of computation required by the task and its own computing power. In general, reasoning on the satellite IoT side can greatly reduce the amount of data that satellites transmit to the ground. At the same time, it saves a lot of valuable network bandwidth resources.

This method is feasible and meaningful. With the development of hardware technology, embedded devices on satellites usually have higher computing power and can meet the computational requirements of common neural network inference. We give an intuitive explanation in Section 4. While meeting the computing needs, satellites also need to consider the issue of energy consumption. In general, the energy consumption of satellites is mainly reflected in the energy consumption of communication and the energy consumption of task processing. Task processing generates a lot of energy consumption. When the inference task is offloaded to the satellite IoT edge node, the satellite IoT edge node faces pressure in terms of computation and energy consumption. Of course, in order to alleviate the pressure on the satellite IoT edge nodes caused by large computing tasks (such as large inference networks such as VGG-16 and WRN), we can choose to put large computing tasks on the satellite IoT cloud nodes. In particular, large computing tasks that do not have timeliness requirements can also be offloaded to the ground data center. In short, the intelligence of satellite IoT edge nodes provides the possibility of real-time processing of edge-side data.

### 3.3. Summary of Satellite IoT Edge Intelligent Computing Architecture

In summary, using the combination of the cross-layer satellite IoT edge intelligent computing architecture and the parameter server algorithm is simpler in the process of gradient sharing. Further, the satellite IoT cloud node cluster can provide highly available support for parameter management. Meanwhile, the architecture has higher requirements on the network, and the communication cost is higher in the process of parameter sharing. Therefore, this architecture is more suitable for intelligent computing tasks with less computation and smaller scale. In contrast, the architecture using the Ring Allreduce algorithm is friendlier in terms of network bandwidth requirements. This is at the expense of a more complex gradient sharing algorithm. Therefore, the architecture is more suitable for the intelligent computing tasks with complex computation.

In general, deploying the IoT edge intelligent computing architecture on a satellite cluster is feasible and meaningful. The intelligent processing on the edge side can greatly reduce the time cost caused by network transmission during the process of converting from original data to valid information, which can realize rapid response and fully utilize the storage and computing capabilities of the satellite nodes. This technology has a very broad development prospect in the application of satellite constellation communication, navigation, and remote-sensing observation.

## 4. Results and Discussions

In order to better evaluate the edge intelligence computing architecture of the satellite IoT, we have established simulation experiments. First, we simulate the connectivity and coverage performance of the satellite IoT. Next, we simulate the training and inference process of the satellite IoT edge intelligent computing architecture, mainly from the aspect of delay.

### 4.1. Satellite IoT Connectivity and Coverage Performance

Unlike the ground-based IoT, the satellite IoT needs to consider high-speed dynamic topology. Fortunately, the satellite IoT can be designed based on satellite constellation, the network topology has

periodic changes, and the motion of the nodes becomes predictable, which also facilitates the analysis of the satellite IoT. Below, we analyze the connectivity and coverage performance of the satellite IoT.

We simulated the connectivity and coverage performance of the satellite IoT through the *STK11.5* (Satellite Tool Kit) simulation software. The constellation of satellites is set to the 66/6/1 Walker constellation, with 66 satellites, 6 orbital planes, and 11 satellites per orbital plane. Each satellite has a height of 1500 km, an orbital inclination of 90°, and a half-cone angle of 50° detected by the sensor. By setting the above parameters in the STK software, we can obtain pictures of different perspectives of the entire satellite IoT, as shown in Figures 5 and 6.

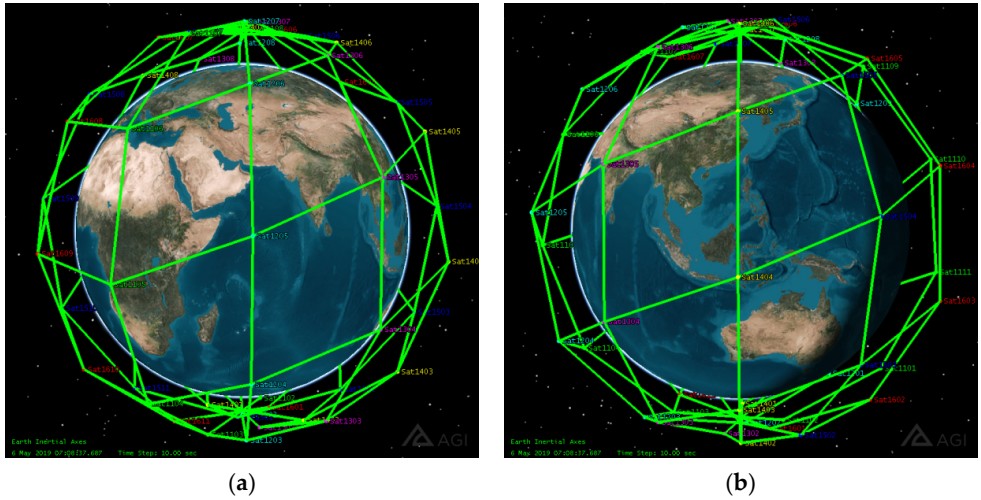

(**a**)                    (**b**)

**Figure 5.** Three-dimensional plan of the satellite Internet of Things. (**a**) Observation point of Sat1205; (**b**) Observation point of Sat1404.

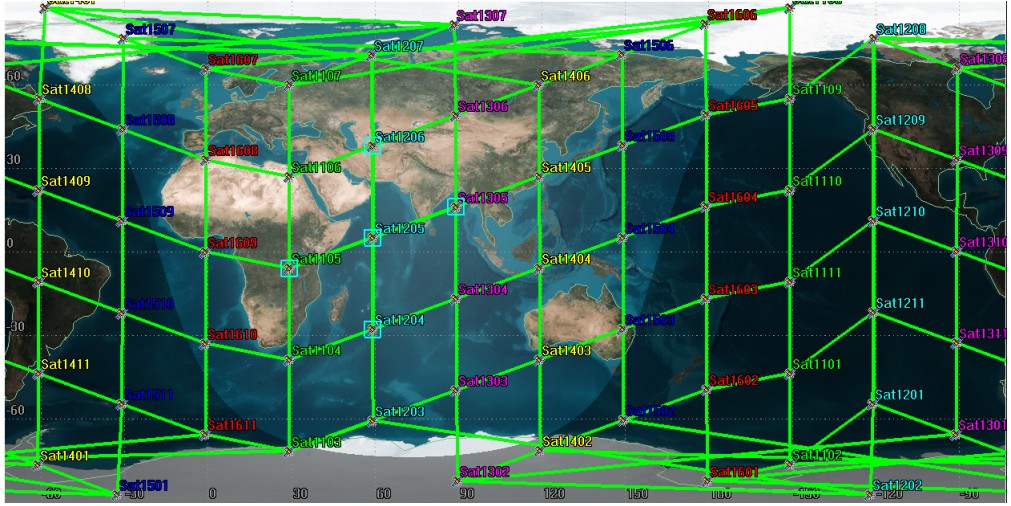

**Figure 6.** Two-dimensional plan of the satellite Internet of Things.

Figure 5 shows the communication connection between each satellite in a real scene. The green line indicates the communication link between the satellite and the satellite. Assuming that a satellite communicates with four azimuth satellites, it can be found that these satellites form a network of multiple quadrilaterals, and the entire satellite Internet of Things is like a network covering the whole world.

Figure 6 shows the projection of the communication link between each satellite in a two-dimensional plan view of the Earth. Due to the curvature of the Earth, under the two-dimensional plan of the Earth, the projection of the communication link between each satellite will be interlaced in the South Pole and the Arctic. The reason for this phenomenon is that, when the three-dimensional sphere is unfolded

into a two-dimensional plane, the graphic display at the two-pole position is elongated, but, in reality, the communication link remains connected.

The STK connection module can calculate the connection data for each satellite. Taking the satellite Sat1205 as an example, through the STK to simulate the trajectory of satellite motion, we observe the link connection of the Sat1205 satellite. By recording the link connection of the Sat1205 satellite, we can see that the Sat1205 satellite can maintain a stable communication link with the surrounding satellites within 24 hours of satellite motion. The connection characteristics of the satellite Sat1205 with the satellites Sat1105, Sat1204, Sat1206, and Sat1305 are shown in Table 1.

**Table 1.** Connection duration and connection distance between satellite and satellite.

| Connection Pair | Connection Duration/Day | Connection Distance/km |
| --- | --- | --- |
| Sat1205-to-Sat1105 | 24 h | 1446.81–4327.11 [1] |
| Sat1205-to-Sat1204 | 24 h | 4439.16 |
| Sat1205-to-Sat1206 | 24 h | 4439.16 |
| Sat1205-to-Sat1305 | 24 h | 1446.81–4327.11 [1] |

[1] It takes about one hour from the minimum to the maximum and then to the minimum.

As can be seen from the table, each satellite can establish a link with four adjacent satellites for a long time, and the link distance between two adjacent satellites in the same orbit remains unchanged (satellite Sat1205 has the same orbital surface as satellite Sat1204 and Sat1206). The link distance between satellites on adjacent orbital planes changes periodically.

Next, we analyze the coverage performance of the satellite IoT. We simulated the global coverage performance of the satellite IoT through STK and calculated it at a point granularity of 3 deg. The percentage of global satellite IoT coverage and the ratio of different latitude coverage times during the day were recorded. Figure 7 shows the coverage characteristics of the entire satellite IoT sensor. By observing the coverage of the satellite IoT sensor in Figure 7, and using STK to calculate the coverage parameters, we record the coverage characteristics of the satellite IoT in Table 2. It can be found from Table 2 that the installed satellite IoT cumulative coverage rate is 100%, and the coverage time of different latitudes also reaches 100%.

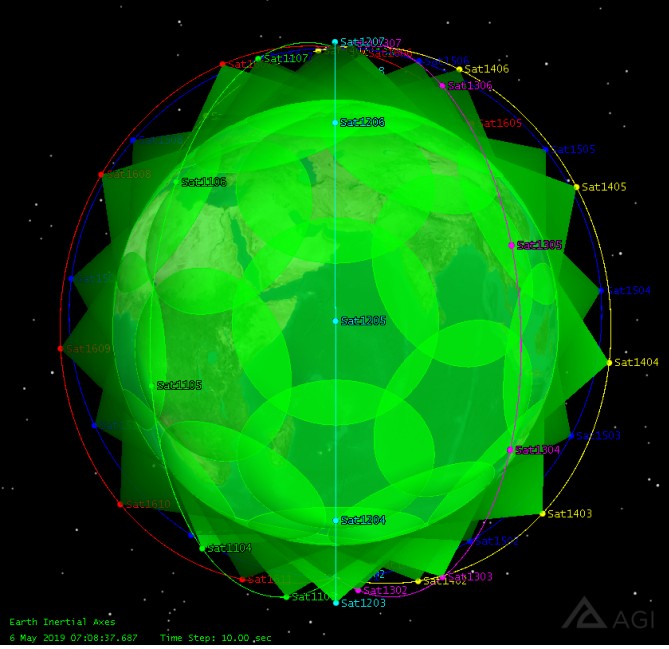

**Figure 7.** The coverage characteristics of the entire satellite IoT sensor.

**Table 2.** Cumulative coverage and latitudes coverage.

| Percent Coverage (Global) | Covering Latitude | Coverage Time of Different Latitudes |
|---|---|---|
| 100% | −90° to 90° | 100% |

We can see that the network topology of the satellite IoT is a periodic stable motion. By designing a reasonable satellite constellation, the connection and coverage performance of the satellite IoT can be improved to some extent.

### 4.2. Satellite IoT Edge Intelligent Computing Architecture Performance

We built a simulation environment in *Cloudsim4.0* [36,37] (cloud computing simulation platform) to evaluate the training and reasoning performance of the satellite edge intelligent computing architecture. The deep-learning training and inference tasks are abstracted into different floating-point operations, and we estimate the total amount of calculation required for training based on the literature, as shown in Table 3.

**Table 3.** Different model training and inference parameters.

| Model | Input Size | Training Set Size | Epochs | Flops (One Pass) | Number of Parameters | Total Calculation |
|---|---|---|---|---|---|---|
| VGG-16 [38] | 224*224*3 | 7000 pcs | 50 | 15.470GFLOPS | 138.38M | 5.164PFLOPS |
| ResNet-50 [39] | 224*224*3 | 7000 pcs | 50 | 3.870GFLOPS | 25.609M | 1.292PFLOPS |
| WRN (wide residual network) [40] | 224*224*3 | 7000 pcs | 50 | 10.935GFLOPS | 68.950M | 3.650PFLOPS |
| MobileNet [41] | 224*224*3 | 7000 pcs | 50 | 0.573GFLOPS | 4.253M | 0.191PFLOPS |
| ShuffleNet [42] | 224*224*3 | 7000 pcs | 50 | 0.136GFLOPS | 1.74M | 0.045PFLOPS |
| DenseNet [43] | 224*224*3 | 7000 pcs | 50 | 2.834GFLOPS | 7.894M | 0.946PFLOPS |

We assume that the inter-satellite link bandwidth is 300 Mbps, and the satellite-terrestrial link bandwidth is 600 Mbps. At the same time, the resource situation of each satellite can be characterized as the parameters of Table 4.

**Table 4.** The resource situation of each satellite.

| | |
|---|---|
| **Floating point computation** | 5 TFLOPS (FP16) |
| **Operating system / Architecture** | Linux/X64 |
| **Virtual machine monitor** | XEN |
| **RAM** | 16GB 256 bit LPDDR4x |
| **RAM Bandwidth** | 2133MHz - 137GB/s |
| **Disk** | 10T |
| **Power** | 30W |

The simulation includes processing, queuing, transmission, and propagation delay, and the propagation speed is the speed of light. Therefore, the end-to-end delay can be expressed as the following:

$$D_{Process} = D_P + D_T + D_L + D_Q$$

Among them, $D_P$ is the processing delay of the task, $D_T$ is the propagation delay, $D_L$ is the data transmission delay and $D_Q$ stands for queuing delay.

During the training process, the input data is the image of 224*224*3 in Table 3. The number of images trained is 7000, and the number of epochs is 50. The output data is a trained parameter, and the size of the data is the size (bytes) of the value of "Number of parameters". Among them, centralized training is expressed as single-node training. The process of distributed training can be expressed as the dispersion of training tasks from one satellite node to other satellite nodes. The transmission and synchronization of parameter data is involved in the distributed training process. During the training process, we will simulate the training duration of different training models on a single node and on multiple nodes, as shown in Figure 8.

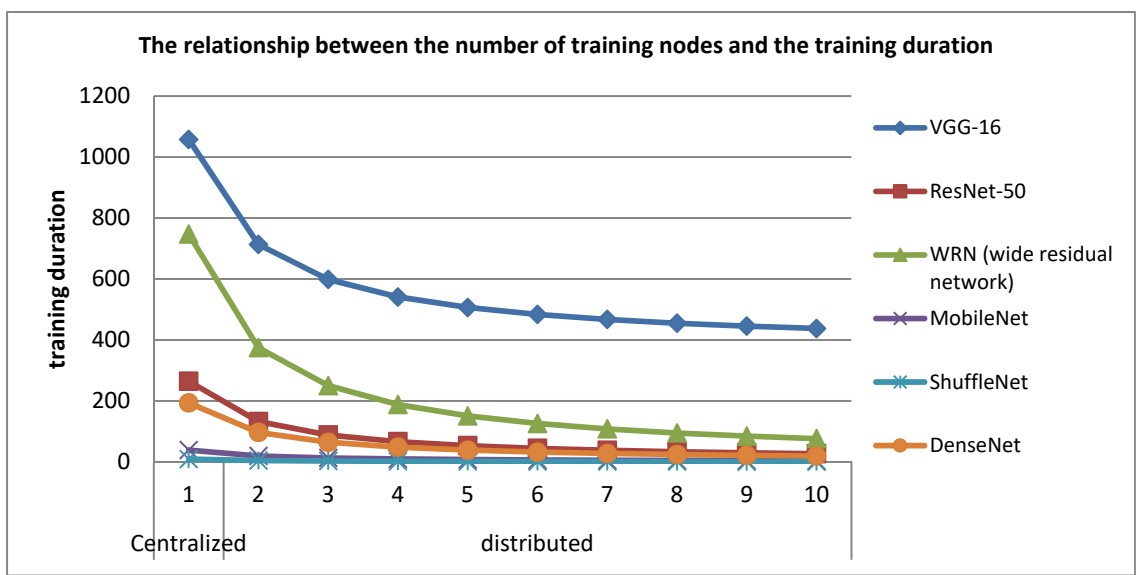

**Figure 8.** The relationship between the number of training nodes and the training duration.

It can be seen from Figure 6 that the environment of the satellite IoT distributed network can effectively reduce the training time of the neural network with relatively large computational demand. For neural networks such as MobileNet and ShuffleNet, which have relatively small computational demands, the rate of change in training time is not obvious, but the overall training time is maintained in a relatively low range. In general, distributed training has a significant effect compared to single-node centralized training. As the number of nodes increases, the training time tends to be flat. When the number of nodes exceeds a certain amount, the training time will not be significantly reduced. In the satellite Internet of Things environment, the increase of distributed nodes will increase the communication delay.

In the reasoning process, the input data is an image of 224 * 224 * 3, and each task is described as processing an image. The amount of computation required to process an image is the value of "Flops (One pass)" in Table 3. In reality, the pixels of the picture produced by the satellite sensor will exceed the image pixels input by the inference process. Therefore, the image of the satellite is divided into several pictures as input data for reasoning. In general, satellite nodes can make inferences locally or share data to nearby satellites for distributed reasoning. In the process of reasoning, we will simulate the reasoning time of a reasoning task and multiple inference tasks under different inference models. First, in the case of a single node, the completion time of different task amounts is as shown in Figure 9.

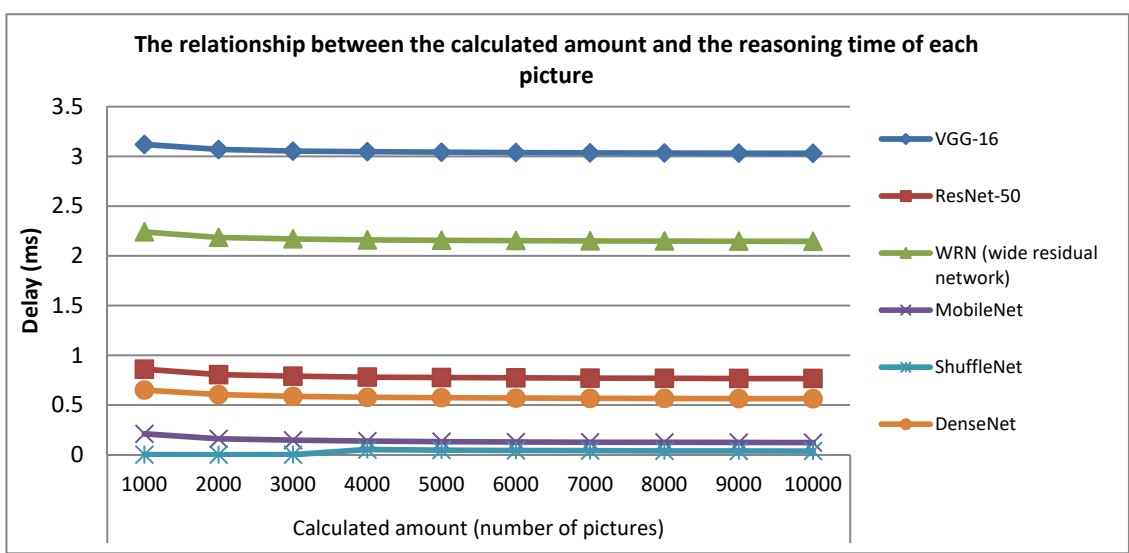

**Figure 9.** The relationship between the calculated amount and the reasoning time of each picture.

As can be seen from Figure 9, in the case of single node inference, when the amount of inference task increases, that is, the number of images to be inferred increases, the inference time of a single picture remains substantially unchanged. The time it takes to complete the reasoning is at the millisecond level. Lightweight neural networks (such as MobileNet and ShuffleNet) have lower inference delays. However, the reasoning delay of the larger neural network model, such as VGG16, is slightly higher.

Next, we will simulate the average inference delay of a single picture of different neural networks in the case of multi-node distributed reasoning, as shown in Figure 10.

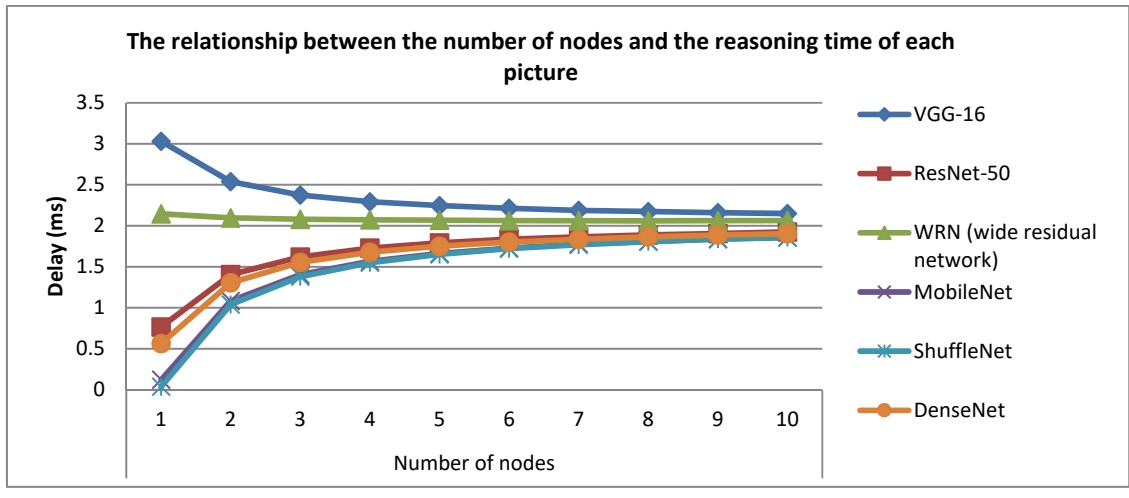

**Figure 10.** The relationship between the number of nodes and the reasoning time of each picture.

Figure 10 shows the effect of the number of different satellite nodes on the inference time of each picture when the calculated amount is 10000. We can find an interesting phenomenon. When the number of satellite nodes participating in reasoning increases, the delay of single-picture inference in some neural network models is decreasing (such as VGG16). However, in some neural network models, the delay of single picture inference is increasing (such as MobileNet and ShuffleNet). Since there are thousands of kilometers of link distance between satellite nodes, and the inter-satellite link bandwidth is not very high, the communication propagation delay and transmission delay between satellites become non-negligible. Especially in the case of small tasks (such as the reasoning of MobileNet and ShuffleNet), the time for satellite single node to make inference is small. If distributed processing is still performed, it will undoubtedly increase the communication delay. For neural networks with

large computational complexity (such as VGG16), distributed reasoning is helpful to speed up the completion of reasoning.

In summary, we can find that lightweight neural networks like MobileNet and ShuffleNet are more suitable for satellite IoT scenarios. In the satellite IoT edge intelligent computing architecture, the training phase of this kind of neural network can be combined with multi-satellite for online distributed training based on the pretraining model, while the inference phase is directly performed locally.

## 5. Conclusions

Satellites have become an important part of the Internet of Things. This paper analyzes the current status and future development trends of satellite IoT networks. Aiming at the problem of satellite IoT data processing, this paper proposes the satellite IoT edge intelligent computing architecture based on the latest development of edge computing and deep learning. Among them, the satellite IoT edge computing and the distributed satellite IoT intelligent computing architecture are described in detail. Finally, this paper simulates and analyzes the connectivity and coverage performance of the satellite IoT and the performance of the satellite IoT edge intelligent computing architecture. By showing the performance of different neural network models in the satellite edge intelligent computing architecture, we can find that the lightweight neural network model can be more suitable for satellite IoT scenarios. The lightweight model of neural networks can promote the development of satellite IoT edge intelligent computing architecture.

**Author Contributions:** Conceptualization, J.W. and S.C.; methodology, J.W., J.H. and S.C.; validation, J.W.; investigation, J.W. and J.H.; writing—original draft preparation, J.W. and J.H.; writing—review and editing, J.W. and J.H.; visualization, J.W. and J.H.; supervision, S.C.; project administration, S.C.

**Funding:** The project was supported by the National Natural Science Foundation of China (No. 617011484), the Research Fund of the manned space engineering (No. 18022010301), the Open Fund of IPOC (BUPT), and the National Defense Science and Technology Innovation Zone Project of China.

**Conflicts of Interest:** The authors declare no conflict of interest.

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
