# Peer review of "Satellite IoT Edge Intelligent Computing: A Research on Architecture"

_electronics, doi:10.3390/electronics8111247_

Round 1

Reviewer 1 Report

Overall, the paper is excellent and present interest information of edge computing.

I'm curious how to calculate energy consumption at each edge, without a master-slave. In the paper, envision offloading the inference task to the satellite IoT edge node, which gives the load on the satellite IoT edge. I think a more detailed explanation of this part should go in.

Author Response

Response to Reviewer 1 Comments

Point 1: I'm curious how to calculate energy consumption at each edge, without a master-slave. In the paper, envision offloading the inference task to the satellite IoT edge node, which gives the load on the satellite IoT edge.

Response 1: Thanks for the careful review, which is very helpful for our work. And we have added a comment about this issue in Section 3.2.2 of the manuscript.

In general, the energy consumption of satellites is mainly reflected in the energy consumption of communication and the energy consumption of task processing. Task processing will generate a lot of energy consumption.

When the inference task is offloaded to the satellite IoT edge node, the satellite IoT edge node will face pressure in terms of computation and energy consumption. Of course, in order to alleviate the pressure on the satellite IoT edge nodes caused by large computing tasks (such as large inference networks such as VGG-16 and WRN), we can choose to put large computing tasks on the satellite IoT cloud nodes. In particular, large computing tasks that do not have timeliness requirements can also be offloaded to the ground data centre.

Reviewer 2 Report

It is a very interesting article and very well written. I have doubts about the future of space scrap, but I think satellite communications will be useful for critical IoT applications. Congratulations

Author Response

Response to Reviewer 2 Comments

Point 1: I have doubts about the future of space scrap, but I think satellite communications will be useful for critical IoT applications.

Response 1: Thank you for your comments and open questions. As the number of satellites increases, the probability of space debris is greatly increased. This is an inevitable problem and a global problem. Of course, we believe that the satellite Internet of Things will have a certain role in promoting space debris processing. In the future, satellite IoT can use sensors to sense and mark space debris around satellite nodes and notify space debris-cleaning aircraft to collect debris.

Reviewer 3 Report

The authors proposed a satellite IoT edge intelligent computing architecture based on the latest development of edge computing and deep learning. Based on the description of the architecture, the authors simulate via CloudSim and Satellite Tool Kit, the performance of the satellite IoT based on the connectivity and the coverage, as well as the performance of the satellite IoT edge intelligent computing architecture. This technical content of the article is significant with some interesting ideas. The quality of presentation is also good.

There are some recommended changes to be made such as

The authors mentioned that the coverage and the connectivity of the satellite IoT is 100% and 24hrs respectively. How can you proof it? The authors need to be more explicit with proof Can the authors get a clearer picture of Figure 5

Author Response

Response to Reviewer 3 Comments

Point 1: The authors mentioned that the coverage and the connectivity of the satellite IoT is 100% and 24hrs respectively. How can you proof it? The authors need to be more explicit with proof Can the authors get a clearer picture of Figure 5.

Response 1: We thank this reviewer for this comment very much. We added more simulation pictures and text descriptions in Section 4.1 to provide proof. At the same time, we also explain the reason why the picture 5 is unclear in the original.

By setting the above parameters in the STK software, we can obtain pictures of different perspectives of the entire satellite IoT, as shown in Figure 5 and Figure 6.

Figure 5 shows the communication connection between each satellite in a real scene. The green line indicates the communication link between the satellite and the satellite. Assuming that a satellite communicates with four azimuth satellites, it can be found that these satellites form a network of multiple quadrilaterals, and the entire satellite Internet of Things is like a network covering the whole world.

Figure 6 shows the projection of the communication link between each satellite in a two-dimensional plan view of the Earth. Due to the curvature of the Earth, under the two-dimensional plan of the Earth, the projection of the communication link between each satellite will be interlaced in the South Pole and the Arctic. The reason for this phenomenon is that when the three-dimensional sphere is unfolded into a two-dimensional plane, the graphic display at the two-pole position is elongated, but in reality the communication link remains connected.

connectivity performance

Taking the satellite Sat1205 as an example, through the STK to simulate the trajectory of satellite motion, we observe the link connection of the Sat1205 satellite. By recording the link connection of the Sat1205 satellite, we can see that the Sat1205 satellite can maintain a stable communication link with the surrounding satellites within 24 hours of satellite motion.

coverage performance

We simulated the global coverage performance of the satellite IoT through STK, and calculated it at a point granularity of 3 deg. The percentage of global satellite IoT coverage and the ratio of different latitude coverage times during the day were recorded. Figure 7 shows the coverage characteristics of the entire satellite IoT sensor. By observing the coverage of the satellite IoT sensor in Figure 7, and using STK to calculate the coverage parameters, we record the coverage characteristics of the satellite IoT in Table 2. It can be found from Table 2 that the installed satellite IoT cumulative coverage rate is 100%, and the coverage time of different latitudes also reaches 100%.

Reviewer 4 Report

The title is about "A Survey on Architecture" but in the paper there are some simulation.

The paper ca not be published in the current form. 

Author Response

Response to Reviewer 4 Comments

Point 1: The title is about "A Survey on Architecture" but in the paper there are some simulations.

Response 1: Thank you for this very nice suggestion. You let us notice that we have a problem with the use of the title word. Combined with the content of the article, our article wants to express the research on the satellite network architecture, not the review article. Therefore, we have made appropriate changes to the title of the article, replacing "Survey" with "Research".

The revised title is: "Satellite IoT Edge Intelligent Computing: A Research on Architecture."

Round 2

Reviewer 4 Report

In my opinion, the paper can be published in current form.